# Brain stimulation competes with ongoing oscillations for control of spike timing in the primate brain

**Matthew R. Krause**[1]◉*, **Pedro G. Vieira**[1]◉, **Jean-Philippe Thivierge**[2,3], **Christopher C. Pack**[1]*

**1** Department of Neurology and Neurosurgery, Montreal Neurological Institute, McGill University, Montreal, Quebec, Canada, **2** School of Psychology, University of Ottawa, Ottawa, Ontario, Canada, **3** Brain and Mind Research Institute University of Ottawa, Ottawa, Ontario, Canada

◉ These authors contributed equally to this work.
* matthew.krause@mcgill.ca (MRK); christopher.pack@mcgill.ca (CCP)

**Data Availability Statement:** All data are available from the Open Science Foundation database: https://osf.io/9t2yp/.

**Funding:** This work was supported by a Canadian Institutes of Health Research Grant to CCP (MOP-

## Abstract

Transcranial alternating current stimulation (tACS) is a popular method for modulating brain activity noninvasively. In particular, tACS is often used as a targeted intervention that enhances a neural oscillation at a specific frequency to affect a particular behavior. However, these interventions often yield highly variable results. Here, we provide a potential explanation for this variability: tACS competes with the brain's ongoing oscillations. Using neural recordings from alert nonhuman primates, we find that when neural firing is independent of ongoing brain oscillations, tACS readily entrains spiking activity, but when neurons are strongly entrained to ongoing oscillations, tACS often causes a decrease in entrainment instead. Consequently, tACS can yield categorically different results on neural activity, even when the stimulation protocol is fixed. Mathematical analysis suggests that this competition is likely to occur under many experimental conditions. Attempting to impose an external rhythm on the brain may therefore often yield precisely the opposite effect.

## Introduction

Transcranial electrical stimulation (tES) is a family of techniques that seek to modulate brain activity by applying electrical current noninvasively, through the scalp. This current flows through the head, producing electric fields that interact with the brain's own electrical activity. Researchers often attempt to target brain functions that rely on oscillatory brain activity at a specific frequency by using transcranial alternating current stimulation (tACS), a form of tES that uses sinusoidal alternating current, oscillating at the same frequency (e.g., [1,2]). Although there have been some concerns about its effectiveness, there is now strong evidence that tACS can influence oscillatory neural activity in vitro [3,4], in small animal models [5,6], and even in the large, well-insulated primate brain [7–9].

Nevertheless, harnessing this mechanism to produce reliable changes in human behavior has proven to be surprisingly difficult. Studies often find that tACS produces inconsistent effects, between and within participants (e.g., [10–12]), even with stimulation frequencies that

115178; https://cihr-irsc.gc.ca), a Parkinson Canada Pilot Project Grant to MRK (PPG-2020-0000000033; https://www.parkinson.ca), and a Natural Sciences and Engineering Council of Canada Discovery Grant to JPT (#210977; https://www.nserc-crsng.gc.ca). The funders had no role in study design, data collection and analysis, decision to publish, or preparation of the manuscript.

**Competing interests:** The authors have declared that no competing interests exist.

**Abbreviations:** CT, computed tomography; DBS, deep brain stimulation; EEG, electroencephalography; LFP, local field potential; MRI, magnetic resonance imaging; PCA, principal component analysis; PLV, phase-locking value; PPC, pairwise phase consistency; tACS, transcranial alternating current stimulation; tES, transcranial electrical stimulation.

are known to be linked to the specific behaviors under study. Although individual differences in neuroanatomy may explain some of these inconsistencies [11], variability in the participants' ongoing brain activity also appears to shape the effects of tACS [13–17]. For example, asking participants to close their eyes—which increases the amplitude of endogenous alpha oscillations—reduces the subsequent effects of tACS [16,17]. In contrast, increasing beta power, by imagining specific movements, seems to increase the effectiveness of tACS [15]. The nature of the interactions between tACS and ongoing brain activity thus remains unclear. Since ongoing brain activity varies between [18,19] and within [20,21] individuals, understanding these interactions is critical for determining when, how, and for whom tACS will be effective.

Most behavioral studies are currently based on the assumption that tACS enhances ongoing oscillatory activity [22], by causing neurons to fire in sync with the stimulation. Indeed, tACS is capable of creating subthreshold membrane potential fluctuations at the stimulation frequency [23,24], which might explain some reports that tACS and ongoing brain activity interact synergistically [15]. At the same time, it has been argued that the relatively weak influence of tACS is likely to be overwhelmed by ongoing neural activity [25–27]. If this were the case, tACS may be unable to alter the activity of neurons that are already entrained to an ongoing oscillation. These hypotheses can best be distinguished by measuring the influence of tACS on individual neurons with varying levels of entrainment to ongoing activity. However, prior neurophysiological experiments, including our own [7–9], have focused on conditions in which ongoing neuronal entrainment was weak. As a result, these experiments did not completely capture the conditions occurring during typical human tACS experiments.

Here, we characterize the interaction between ongoing oscillations and tACS by recording single-neuron activity in the nonhuman primate brain. Our results confirm that tACS can entrain neural activity, but we find that this occurs only when spike entrainment to ongoing activity is weak. Surprisingly, when neurons are strongly locked to ongoing activity, applying tACS usually leads to a decrease in entrainment, which can only be reversed at higher stimulation amplitudes. Since the effects of tACS vary categorically with the strength of ongoing neural entrainment, our data indicate that it competes with brain oscillations for control over spiking activity. Moreover, we show that this competition is a straightforward mathematical consequence of interactions between oscillators.

These results have important ramifications for neuromodulation applications. On the one hand, the reduction of spike entrainment that we observe is precisely the opposite of the effect intended in most studies of human participants. On the other hand, we suggest that the same mechanism could be useful in attaining specific behavioral or clinical goals that require targeted desynchronization of neural activity. More generally, these findings also offer a possible mechanistic explanation for the extensive variability reported in the human tACS literature.

## Results

We examined the interplay of ongoing oscillations and tACS using recordings from nonhuman primates (*Macaca mulatta*), a model system that captures many aspects of human anatomy, physiology, and tACS use. Animals were trained to perform a simple visual fixation task that minimized sensory and cognitive factors that influence oscillations. As animals performed this task, we recorded single-unit activity using standard neurophysiological techniques and assessed neuronal entrainment to ongoing local field potential (LFP) and tACS oscillations.

Our experiments targeted cortical area V4, where neurons often exhibit reliable entrainment to the LFP, especially in the 3 to 7 Hz "theta" frequency band [28]. We verified that this occurred in our experiments by computing phase-locking values (PLVs) that describe the

consistency of spike timing (see Materials and methods). These values range from 0 (spiking occurs randomly across an oscillation's cycle) to 1 (spiking occurs at only a single phase of the oscillation). We first assessed V4 neurons' entrainment to ongoing oscillations by computing PLVs summarizing neurons' entrainment to LFP components between 1 and 100 Hz (in ±1 Hz bins). Under baseline conditions without any tACS, many V4 neurons were locked to the approximately 5 Hz component of the V4 LFP; entrainment to higher frequency components was much weaker and often approached zero (S1 Fig). Even at 5 Hz, the strength of entrainment ranged from 0 to 0.42, providing an ideal way to test how tACS influences neural entrainment across different levels of entrainment conditions.

## tACS causes bidirectional changes in spike timing

Next, we recorded from the same V4 neurons during the application of tACS at 5 Hz. Our stimulation methods closely mimicked those commonly used in human studies and produced an electric field of similar strength (approximately 1 V/m). The effects of tACS were assessed by comparing PLVs obtained during blocks of tACS against those computed from intervals of baseline or "sham" stimulation that were randomly interleaved as a control.

Data from 4 example neurons are shown in Fig 1A, which summarizes the phases at which spikes occurred during the baseline (blue) and tACS (orange) conditions. For these cells, the entrainment to the ongoing oscillation was weak, and the application of tACS led to a significant increase in phase locking ($p < 0.05$, per-cell randomization tests; see Materials and methods). These effects resemble those reported previously in other brain regions [7–9], but only occurred in 11% (17/157) of the neurons from which we collected data.

Instead, the application of tACS led to a decrease in the phase locking of many more V4 neurons, as shown in the 2 example cells of Fig 1B, whose spike times became more uniform during stimulation. Statistically significant decreases in entrainment ($p < 0.05$; per-cell randomization tests) were found in 30% (47/157) of the neurons in our sample. Indeed, decreased entrainment was the predominant influence of 5 Hz tACS at the population level: The median PLV decreased from 0.066 (95% CI: [0.044 to 0.085]) to 0.031 [0.007 to 0.058], a statistically significant reduction in rhythmicity ($p < 0.01$; $Z = 3.86$; Wilcoxon sign-rank test). These changes were not accompanied by changes in firing rate: The median firing rate under baseline conditions was 4.6 Hz (95% CI: [3.7 to 5.8]) and 3.9 Hz (95% CI: [3.4 to 5.4]) during tACS, which was not significantly different ($p > 0.1$; $Z = 1.63$; Wilcoxon sign-rank test). This suggests that these changes in entrainment were not due to signal loss or other artifacts [8]. In fact, when multiple neurons were recorded simultaneously, we often observed that tACS increased the PLV of one neuron, while decreasing the PLV of neurons recorded on an adjacent channel (150 μm apart).

Neurons that had higher levels of baseline entrainment tended to become less entrained during tACS, as can be seen in the example cells of Fig 1A and 1B. This pattern was evident in our complete data set (Fig 1C), where baseline entrainment was significantly and negatively correlated with the subsequent changes during tACS ($\rho = -0.38$; $p < 0.01$). This correlation persisted even after the application of Oldham's method [29] to guard against regression to the mean; a permutation-based analysis [30] yielded similar results.

## Decreased entrainment is not specific to stimulation frequency or brain region

The decreased entrainment we have observed may seem at odds with results from previous studies, which have consistently reported increased entrainment with tACS for spikes recorded from different brain regions and with different tACS frequencies [5–9]. We therefore asked

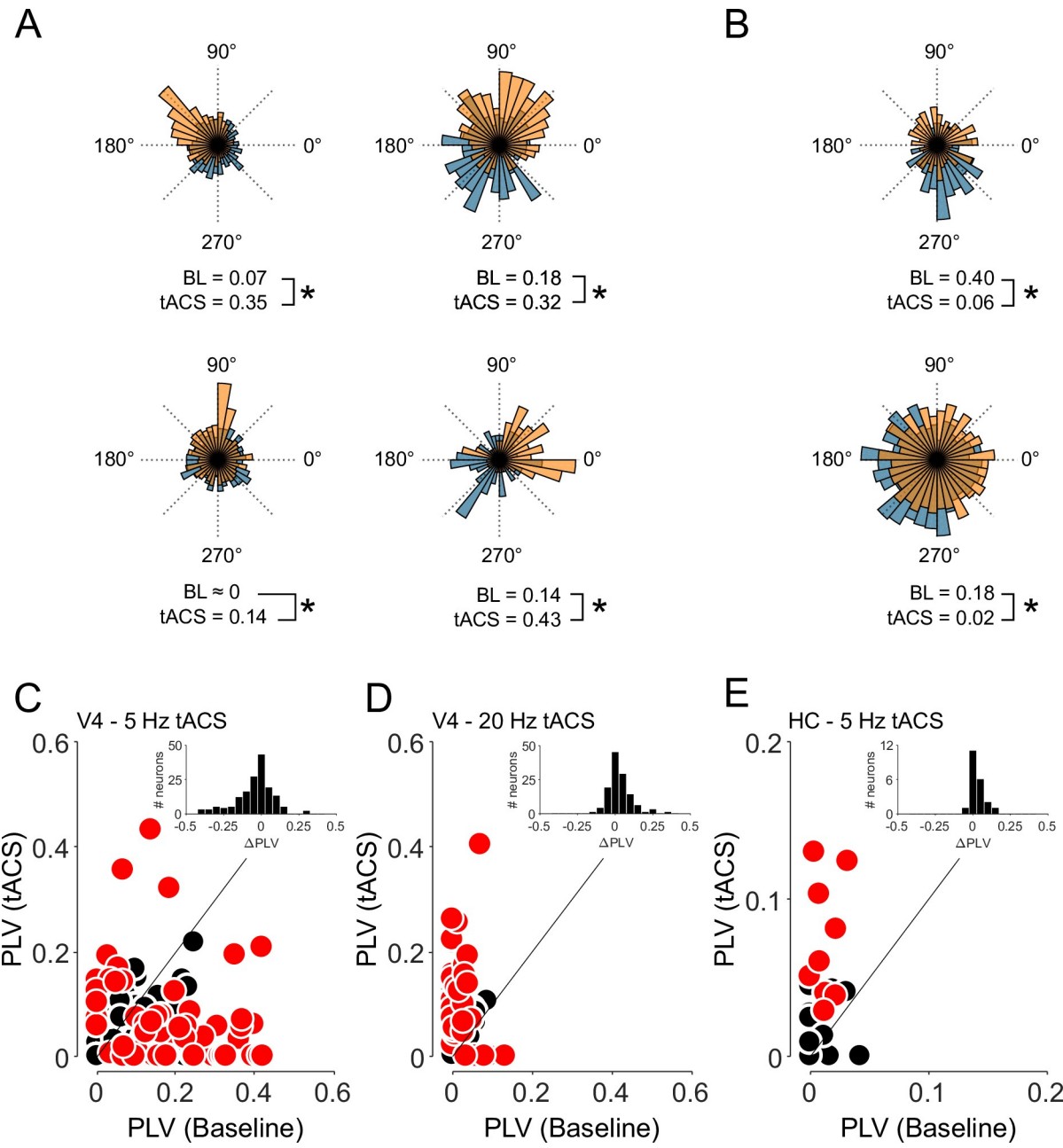

**Fig 1. Applying tACS during physiological oscillations results in bidirectional changes in spike timing. (A, B)** Spike density histograms for 6 example V4 neurons showing the relative amounts of spiking across an oscillatory cycle. The cells in (A) showed increased entrainment during tACS (orange), compared to baseline (blue). This was sometimes (right column), but not always (left) accompanied by a shift in preferred spiking phase. However, many more neurons, like those shown in (B), had decreased entrainment during 5 Hz tACS compared to baseline. PLV values for each condition are shown below; asterisks indicate $p < 0.01$. **(C–E)** Each point in the scatter plots represents a neuron's PLV during baseline (horizontal position) and tACS (vertical). Data were collected (C) in the presence of a 5 Hz ongoing oscillation within V4 ($N = 157$), (D) also in V4, but at 20 Hz, where the oscillation is weaker ($N = 123$), and (E) again at 5 Hz but in the hippocampus (HC; $N = 21$), which also lacks strong 5 Hz oscillations under our conditions. Neurons showing individually significant changes in phase locking are denoted in red ($p < 0.05$; per-cell randomization tests). Inset histograms show the changes in PLV across the population. Moreover, 28/123 cells in panel D were also reported in (9); panel E adapted from (8). See also S1 Fig for comparison of entrainment during baseline conditions and S1 Data for numeric values. BL, baseline; PLV, phase-locking value; tACS, transcranial alternating current stimulation.

whether the lack of entrainment was due to our choice of brain region or stimulation frequency.

As a within-area control, we applied 20 Hz tACS instead (Fig 1D). Neurons in V4 were weakly entrained to the 20 Hz LFP component under baseline conditions (median: 0.01; 95% CI: [0 to 0.018]), with only 1/123 neurons having a PLV above 0.1. Applying 20 Hz tACS to V4 caused 22% (27/123) of neurons to fire significantly more rhythmically ($p < 0.05$; per-cell randomization test; Fig 1D). The population PLV was significantly increased ($p < 0.01$, Z = −3.945; Wilcoxon sign-rank test), tripling from 0.01 to 0.029 [0 to 0.050]. These data demonstrate that V4 neurons can be entrained by tACS at a different frequency, suggesting that the 5 Hz results are not attributable to brain area.

As a within-frequency control, we also examined neural entrainment in the hippocampus, where baseline entrainment at 5 Hz was weak (Fig 1E). Here, the median baseline PLV was 0.01, and none of the 21 neurons in our sample had a 5 Hz PLV above 0.1. Applying 5 Hz tACS increased entrainment in 9 of 21 hippocampal neurons (42%), with the median PLV rising from 0.0083 [0 to 0.02] under baseline conditions to 0.04 [0.01 to 0.05] during stimulation ($p < 0.01$; Z = -3.06; Wilcoxon sign-rank test; Fig 1E). Together, these results show that decreased entrainment is neither a feature of V4 neurons nor of 5Hz tACS, both of which can be associated with increased entrainment under the right circumstances, as shown in Fig 1D and 1E.

## tACS alters the strength and phase of entrainment

Some cells, such as the examples shown in the right column of Fig 1A, also showed a shift in the oscillatory phase at which spiking most often occurred. The specific timing of spikes within an oscillation (i.e., phase) may encode additional information [31], so shifts in preferred phases represent another dimension along which tACS can alter neural activity. In principle, these changes could occur even when the overall levels of entrainment are not affected by tACS.

We therefore examined the subset of neurons whose PLVs were not significantly affected by tACS to see if their preferred phase of spiking shifted during stimulation. Since phase shifts require neurons to have a preexisting phase preference, we first analyzed the 22/93 neurons that had individually significant phase preferences ($p < 0.05$; Rayleigh tests) under baseline conditions. Fig 2 plots the PLVs and preferred phases jointly for each neuron in each condition. The radial distance from the origin corresponds to a neuron's PLV, while the angular position denotes the phase preference. Vectors connecting the baseline values (blue dots) with those measured during tACS (orange dots) therefore completely describe the effects of stimulation on spike timing. For these neurons, applying tACS consistently shifted the neurons' spiking phase, so that they developed a statistically significant ($p < 0.01$; Hodges–Ajne omnibus test) preference for firing during the rising phase of the tACS waveform (35.6˚, red arrow in Fig 2A).

Repeating this analysis using all 93 neurons where 5Hz tACS had no effect on PLV revealed a similar result: a statistically significant phase-shift toward 38˚ ($p < 0.01$; Hodges–Ajne omnibus test). The direction of these shifts is shown in the violet histogram of Fig 2A. These phase changes cannot be attributed to differences in referencing in the baseline and tACS conditions or to spike waveform distortions caused by tACS (see Materials and methods). In fact, shifts toward this early rising phase of the tACS cycle have been predicted from biophysical properties of neuronal membranes [32].

Similar visualizations for neurons that showed significant changes in PLV are provided in Fig 2B and 2C. Neurons that became entrained by tACS tend to start near the origin under baseline conditions and proceed outward during stimulation (Fig 2B). Likewise, when tACS

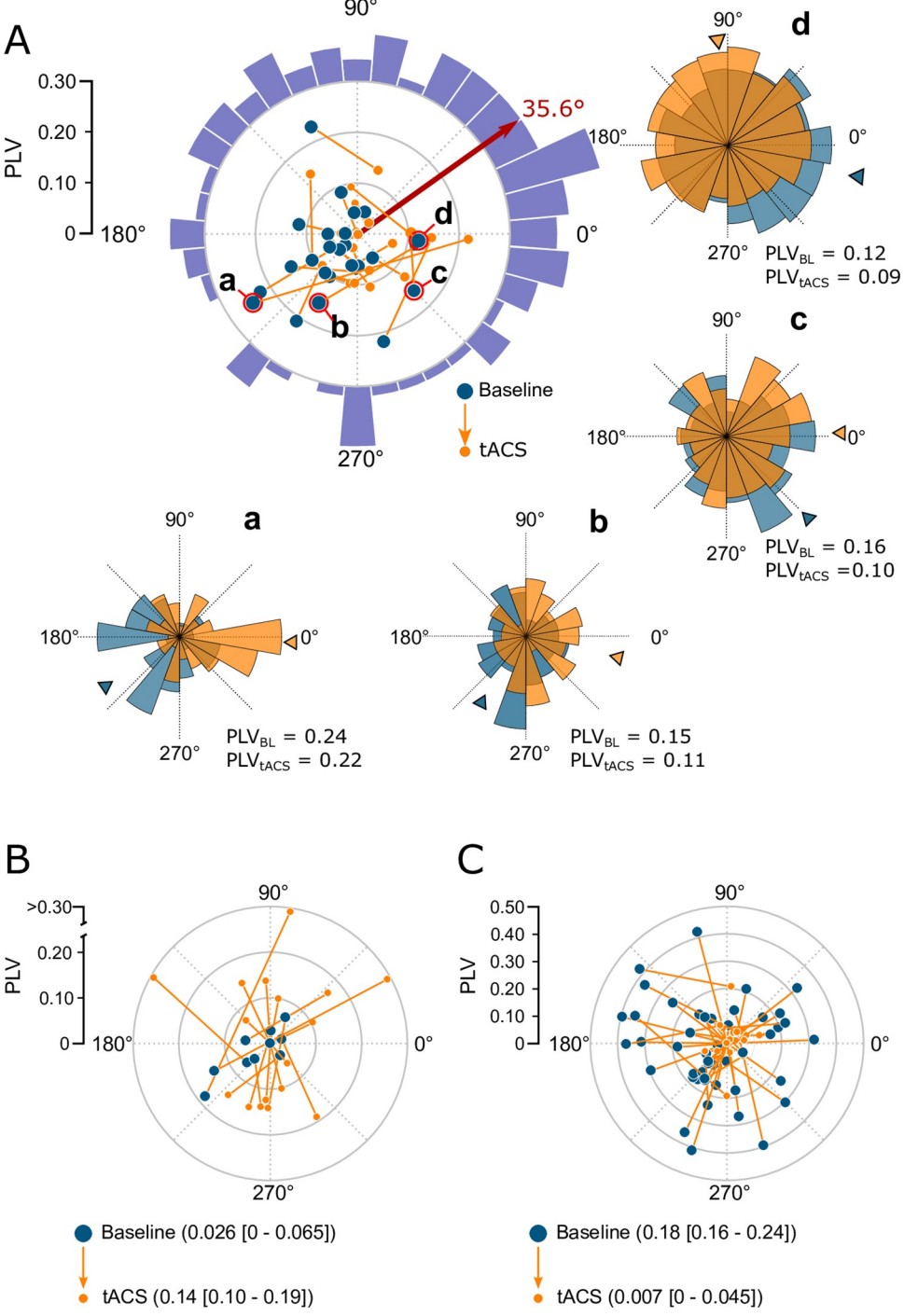

**Fig 2. Ongoing oscillations determine the strength and phase of tACS entrainment.** These polar plots summarize the combined effects of tACS on entrainment strength (PLV, eccentric direction) and entrainment phase (polar angle). Each vector begins during baseline (blue dot) and extends to 1 mA tACS condition (gray dot). **(A)** Individual vectors for the 22 neurons that had no significant change in PLV but a significant phase preference during baseline. The red arrow indicates their average direction of change. Spike phase histograms are shown for the 4 example cells circled in red. The violet histogram shows similar changes in direction across the entire population of 93 neurons whose PLVs were not significantly affected by tACS. It also peaks at 38˚, near the red arrow tip. **(B, C)** Neurons showing individually significant increases (B) and decreases (C) in PLV, plotted in the same style. No red arrows are included in these panels because the population has no net phase preference ($p > 0.05$; Hodges–Ajne omnibus test), in part because neurons had weak phase preferences in either the baseline condition (B) or tACS condition (C), as indicated

by the clustering near the origin. See S1 Data for numeric values. PLV, phase-locking value; tACS, transcranial alternating current stimulation.

reduced a neuron's entrainment to the ongoing oscillation, its spiking activity started in the periphery under baseline conditions and proceeded inwards (Fig 2C). We therefore suggest that the phase shifts observed in the cells where PLV values were not significantly changed reflect a balanced combination of these effects: tACS has decoupled neurons from their entrainment to the ongoing oscillation and imposed a similar amount of synchronization to the stimulus waveform at a new phase.

## Current intensity shifts the dominant effect of tACS

To test this possibility, we collected additional data from 47 V4 neurons using ±1 mA and ±2 mA stimulation. We reasoned that if weak stimulation were partially reducing entrainment to a physiological oscillation, stronger electric fields could completely overcome it and lead to increased entrainment to the tACS waveform [33].

Fig 3 shows the results of this experiment. As in Fig 2, the location of each blue dot represents a neuron's baseline entrainment, in terms of overall PLV (radial distance from the origin) and preferred phase (angle). Spike timing during ±1 mA is depicted by orange dots and ±2 mA by red dots, forming a trajectory that shows the effects of increasing current on spike timing. Our hypothesis predicts that these trajectories should have a specific form: Starting from the neurons' baseline levels of entrainment, they first move toward the origin as the tACS and LFP vie for control of spike timing. Once tACS overwhelms the baseline entrainment, it imposes its own rhythm and the vectors extend outward from origin toward the rising phase of the tACS.

Many neurons exhibited this pattern: Applying ±1 mA tACS decreased or eliminated entrainment relative to baseline, but the stronger ±2 mA stimulation reinstated some entrainment, often at a different phase. The example neuron shown in Fig 3A had a PLV of 0.159 under baseline conditions, no detectable entrainment during ±1 mA tACS (PLV ≈ 0), and a PLV that again reached 0.158 during ± 2 mA stimulation. During this transition from 0 to ±2 mA stimulation, its preferred phase rotated from 191˚ to 83˚. While a naive analysis that considered only changes in PLV might conclude that this neuron was insensitive to tACS, these data demonstrate that the structure of its spike timing is, in fact, altered by the stimulation.

Other neurons progressively lost entrainment as the stimulation amplitude increased. This pattern appears in the example cell shown in Fig 3B, whose PLVs decreased from 0.21 to 0.092, to nearly 0 as the tACS amplitude increased, with the phase consistently remaining near 225˚ (223˚, 235˚, and 221˚, respectively). Both patterns are common in our data, as shown in Fig 3C, which depicts the trajectories through 0, ±1, and ±2 mA for the 28/47 neurons (60%) in our data set that showed significant changes from their baseline PLV at either tACS intensity. One possible interpretation of these data is that neurons vary in their susceptibility to the electric field (e.g., due to their morphology or orientation relative to the field) and that stimulation above ±2 mA would be necessary to completely desynchronize and reentrain the cells in the latter group.

We present a similar diagram for our hippocampal data in Fig 3D. Although the electric field reaching this deep structure was far weaker (0.2 to 0.3 V/m; 8), nearly half of these neurons (9/21; Fig 1E) showed increased entrainment, reminiscent of the ±2 mA condition in V4. Critically, these neurons' baseline levels of entrainment were near zero, so it was not necessary to overcome any substantial baseline entrainment before entraining their spike timing. Interestingly, the preferred phase for these neurons was similar to that observed in V4 (Fig 3C).

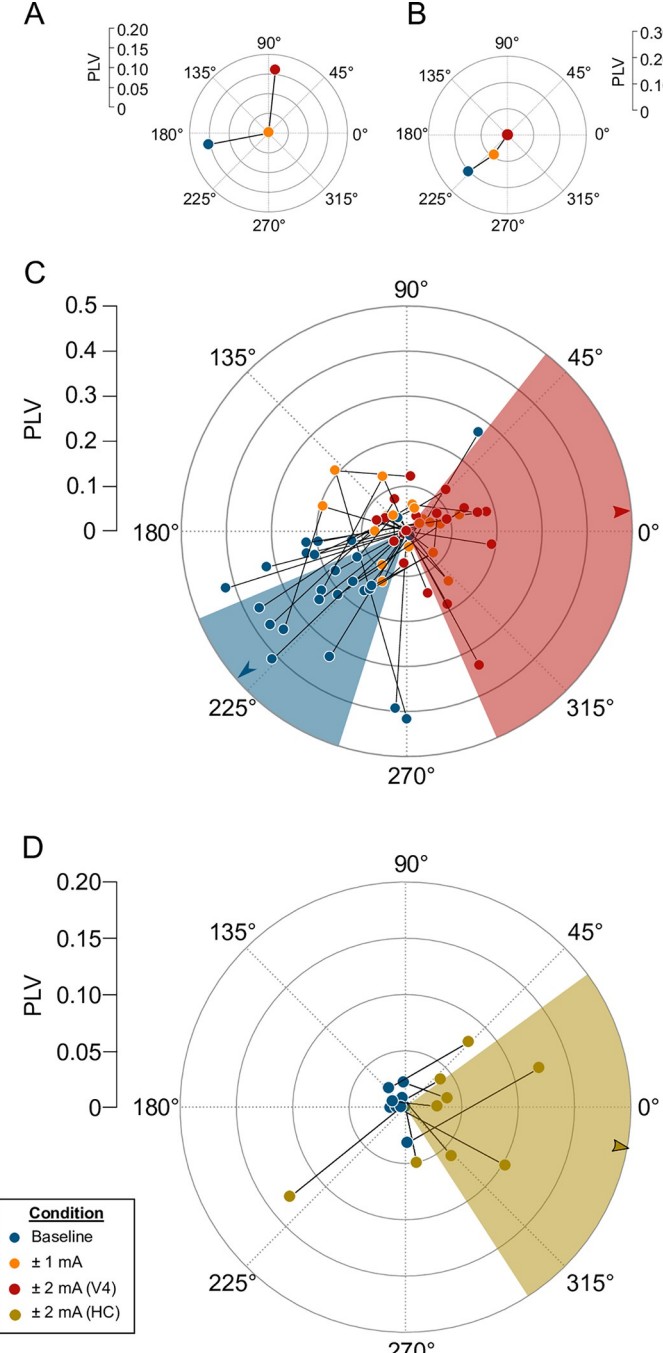

**Fig 3. tACS reduces entrainment, then reinstates it at a different phase, as the stimulation amplitude increases.**
**(A–C)** Polar vectors indicating the phase and strength of V4 neurons' entrainment (as in Fig 3). The dots indicate
baseline conditions (blue), followed by ±1 mA tACS (orange) and ±2 mA tACS (red). Shading indicates the median
and 95% CI of the phase preference during baseline (blue) and ±2 mA tACS (red). Panels A and B show an example
trajectory for one neuron exhibiting reinstated entrainment (A) and another undergoing progressive decreases in
entrainment (B). Panel C shows population data from 27 neurons with significant changes in PLV at either amplitude.
**(D)** Data from $N$ = 9 hippocampal neurons during baseline (blue dot) and ±2 mA tACS (olive), plotted as in panel A.
Note that ±2 mA tACS produces an electrical field within the hippocampus that is 3 times weaker than the ±1 mA
condition for V4. See S1 Data for numeric values. PLV, phase-locking value; tACS, transcranial alternating current
stimulation.

This similarity has been suggested to indicate that both populations were directly affected by tACS via the same biophysical mechanism and that the deep structure was not merely entrained by modifying its more superficial inputs [6]. Overall, these results, across stimulation frequencies, stimulation amplitudes, and brain regions, consistently show that tACS influences spiking activity in a manner that depends strongly on the levels of preexisting entrainment.

## Similar patterns of effects emerge from a simple oscillator model

Our experimental results show a specific pattern where weaker stimulation reduces entrainment relative to baseline, while stronger stimulation successfully entrains neurons. Although there are many ways in which tACS could interact with ongoing oscillations (reviewed in 34), most previous experimental work has not reported decreased entrainment. We therefore sought to determine whether decreased entrainment has a theoretical basis in the properties of coupled oscillators. Specifically, we explored the properties of a simple oscillator model [35] consisting of 2 equations:

$$\frac{dx}{dt} = \lambda x - \omega y - \gamma(x^2 + y^2)x + ks(t)$$

$$\frac{dy}{dt} = \lambda y + \omega x - \gamma(x^2 + y^2)y$$

Here, $x$ and $y$ represent the dynamics of 2 coupled populations of neurons, whose interactions generate an oscillation. For simplicity, we assumed that higher amplitude oscillations were associated with stronger phase locking, as found in previous experimental work [6,24].

We simulated the effects of tACS by changing the properties of the external drive $s(t)$, setting it either to zero (for baseline conditions) or a sine wave of given frequency and phase offset for the tACS condition. The parameter $k$ determines how strongly this external drive affects the neuronal population and is given here as a percentage of the ongoing oscillation's amplitude.

We first asked whether phase mismatches between an ongoing oscillation and tACS at the same frequency could account for our results. They cannot. Most phase offsets do not reduce the oscillation's amplitude, even transiently, but instead lead to a shift in phase with the amplitude preserved or increased. Phase offsets near 180 degrees can produce brief reductions in amplitude, but the oscillation rapidly recovers and eventually exceeds the baseline amplitude. Fig 4A shows the most extreme example of this effect. Applying completely out-of-phase stimulation (180° mismatch) initially depresses the oscillation's amplitude, but these effects only last 3 to 6 cycles; the amplitude subsequently increases by 5%, 26%, and 49%, depending on the stimulation intensity. Since our data was collected in blocks where hundreds or thousands of tACS cycles were applied, any transient decrease in the oscillation would be overwhelmed by the subsequent long-lasting amplitude increase. Moreover, our data were collected during an open-loop stimulation paradigm, where the onset of stimulation was unrelated to the ongoing oscillation's phase, so conditions where even these transient decreases occur were unlikely.

We next asked whether slight differences between the frequencies of tACS and the ongoing oscillation could account for our results. This did seem to be the case. Fig 4B shows that applying stimulation at a frequency that was only 6% below that of the ongoing oscillation led to a 29% decrease in amplitude at moderate stimulation intensities ($k$ = 30%; middle row of Fig 4B), but a 35% increase at higher intensities ($k$ = 75%; bottom row of Fig 4B). Similar effects were observed when stimulation frequency was slightly higher instead: The weaker stimulation decreased the oscillation's amplitude by 25%, while the stronger stimulation increased it by 44% instead. These effects are summarized in Fig 4C for a wide range of stimulation strengths and relative frequencies. To mimic the open-loop nature of our experiments, these data were

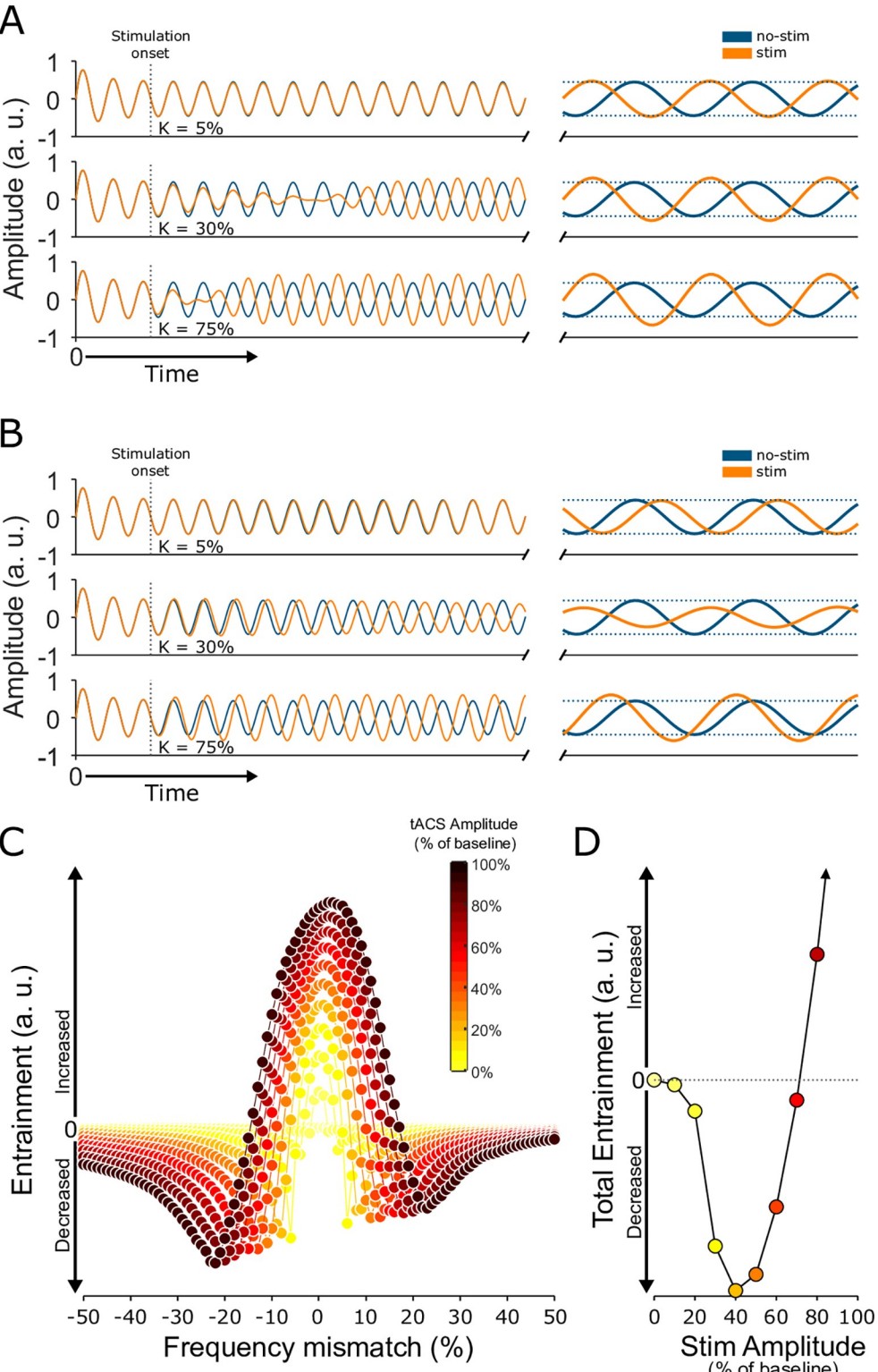

**Fig 4. A simple oscillator model replicates these results. (A)** Stimulating the Stuart–Landau model at the same frequency tends to increase entrainment and shift phase. The model's output, *x(t)*, is shown during a no-stimulation condition (blue) and during the application of tACS (yellow) at low (top), medium (middle), and high (bottom) stimulation intensities. Dotted lines indicate oscillation's amplitude in the absence of stimulation. The other component of the model, *y(t)*, is simply a phase-shifted version of *x(t)* and shows the same pattern of effects, as shown

in S2 Fig. **(B)** Stimulation applied at a slightly different frequency tends to reduce entrainment instead. The model output is plotted as in panel A. **(C)** Changes in entrainment, as a function of frequency mismatch and stimulation intensity. **(D)** Changes in total entrainment, calculated by integrating the curves in panel C. Colors correspond to those used in panel C. See also S1 Appendix and the figures therein. See S2 Data for numeric values. tACS, transcranial alternating current stimulation.

averaged across evenly spaced phase offsets, so these effects are independent of the relative phases of the tACS and ongoing oscillation.

In summary, stimulation that precisely matched the oscillation's frequency, regardless of phase, eventually led to increased oscillation strength but detuning the frequency by even a fraction of a cycle reduced the amplitude noticeably when the stimulation amplitude was relatively weak. In real experiments, such a mismatch is likely unavoidable, given variability in spike timing [36] and peak frequencies within an oscillation [37], as well as the nonsinusoidal nature of neural oscillations [38,39]. We obtained similar results, shown in S1 Appendix, from simulations using Ornstein–Uhlenbeck processes, which mimic the irregular nature of real neural oscillations [40].

Since entrainment is typically measured across a range of frequencies, as in the 2 Hz bins used in our analysis above, tACS at a single frequency cause mixed effects on entrainment. We therefore examined the net effect of stimulation on narrow-band oscillations, by integrating the curves of Fig 4C across frequencies for each stimulation intensity (Fig 4D). Initially, the total entrainment within these narrow frequency bands decreased, but when the stimulation amplitude exceeded about 66% of the ongoing oscillation's amplitude, entrainment increased, just as we have observed in our data (Fig 3).

## Baseline frequency preference determines the effects of tACS

A straightforward prediction of this model is that entrainment should be most strongly reduced for neurons already entrained to frequencies that differ slightly from the stimulation frequency (i.e., in the flank of the entrainment versus frequency mismatch curve in Fig 4C). We tested this prediction in our experimental data by identifying each V4 neuron's baseline frequency preference. To do so, we filtered the baseline LFP into narrow frequency bands between 2 and 8 Hz ($\pm 0.25$ Hz around the center frequency) and calculated the PLV between spiking activity and each LFP component. The center frequency of the component with the highest PLV was assigned as the neuron's preferred frequency, as summarized in Fig 5A.

Fig 5B shows that, of the 47 neurons showing significantly decreased entrainment during 5 Hz tACS, neurons with baseline frequency preferences 1 to 2 Hz away from the stimulation frequency had the largest decrease in PLV. This is reminiscent of the flank of Fig 4C. Only one neuron with decreased entrainment had a baseline frequency preference between 4.75 and 5.25 Hz. Owing to their low baseline entrainment, no preexisting frequency preference could be identified for the 17 neurons that exhibited increased entrainment during tACS.

The model also predicts that stimulation far from the baseline frequency preference should have little effect on entrainment at that frequency. Consistent with this hypothesis, we found little change in entrainment to the 5 Hz component of the LFP during 20 Hz tACS: Only 13/123 or 10.5% of cells had significantly altered PLVs, a percentage that is expected given our $p = 0.05$ threshold for individual cells ($\chi^2(1) = 2.2$; $p = 0.14$). Taken together, these results demonstrate a strong but qualitative agreement with models of competing oscillators.

## Excitatory and inhibitory cells are similarly affected by tACS

The model makes another specific prediction that can be tested in our data. Previous work has suggested that tACS may preferentially target certain types of neurons: Due to their

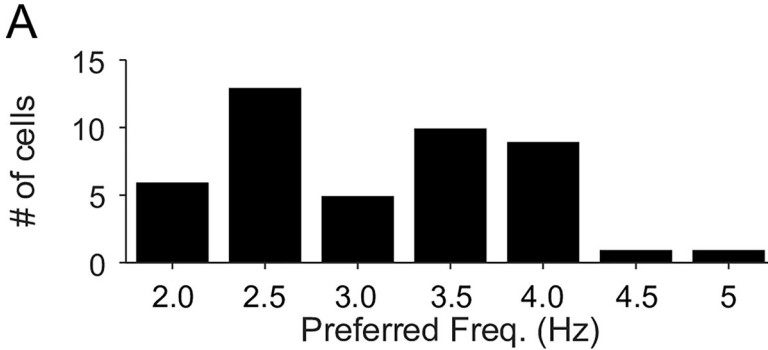

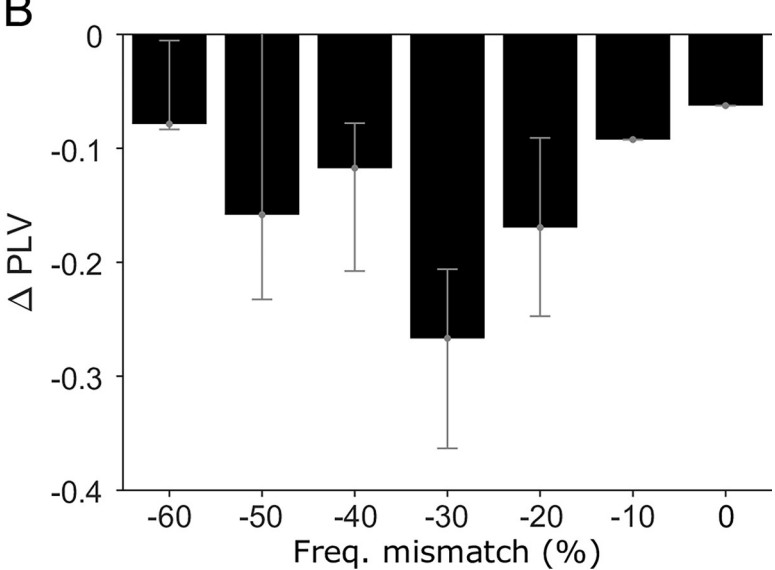

**Fig 5. Preexisting frequency preference determines the effects of tACS. (A)** Histogram of narrowband preferred frequency for the 47 V4 neurons with significantly decreased entrainment during 5 Hz tACS. **(B)** Changes in these neurons' PLV as a function of the difference between their narrowband preferred frequency and the 5 Hz tACS frequency. Note the qualitative similarity to the flank of Fig 4C. See S1 Data for numeric values. PLV, phase-locking value; tACS, transcranial alternating current stimulation.

morphology, interneurons have been proposed to be less susceptible to tACS [41,42]. Others have suggested that, due to their positions within neural circuits, interneurons are actually more strongly affected by tACS [43]. In the model, stimulation is applied to one subpopulation (the $x$ term above), but both subpopulations are affected similarly by the stimulation (S2 Fig), due to the tight coupling between the equations. This suggests that all neurons, regardless of cell type, will be affected similarly by tACS.

We therefore asked whether cell type could explain the mix of increased and decreased entrainment shown in Fig 1C. In our experiments, putative cell types were identified via their spike widths: Narrow-spiking neurons are often interneurons, while those with broader spikes are more likely to be pyramidal cells (Fig 6A; see Materials and methods). As predicted from the model, broad-spiking neurons in our data were no more likely than narrow-spiking neurons to be significantly modulated, in either direction, by tACS ($p = 0.44$; $X^2$ (1) = 0.59). When considering only the neurons affected by tACS (Fig 6B, colored dots), the 2 cell types did exhibit significantly different directions of modulation ($p < 0.01$; $X^2(1) = 12.1$). Entrainment

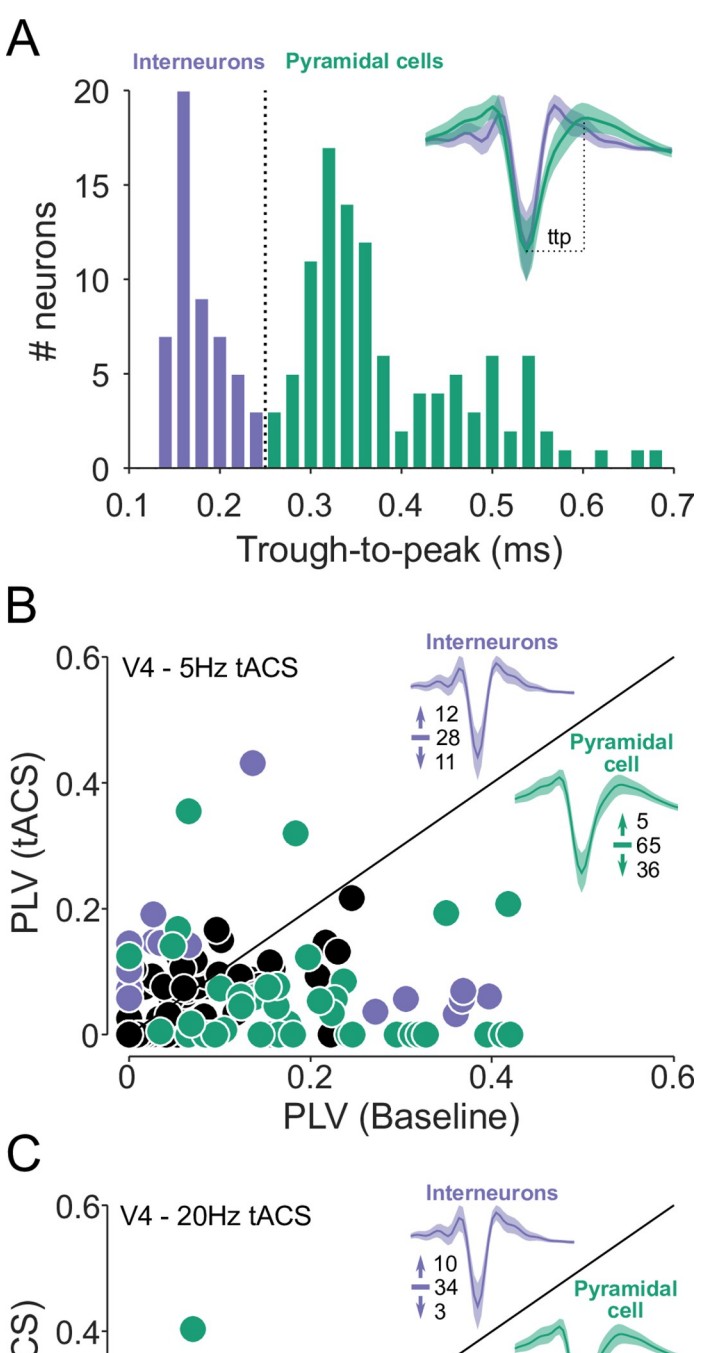

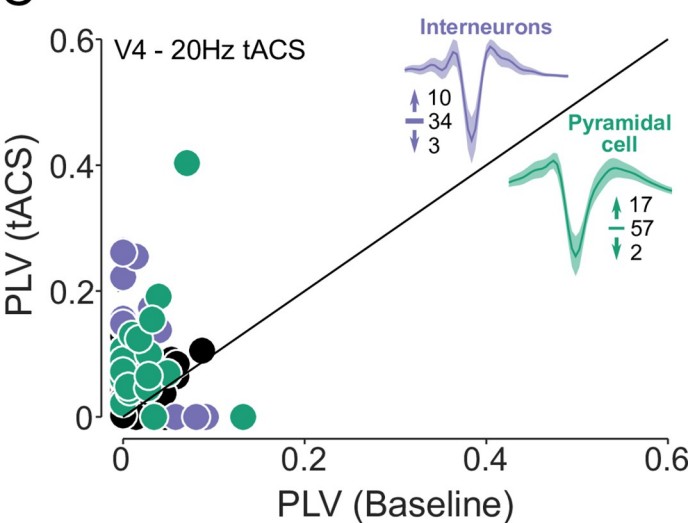

**Fig 6. Distinct cell types are associated with tACS synchronization versus desynchronization. (A)** Neurons ($N = 157$) were divided using a trough-to-peak width threshold of 250 μs (dashed line). Narrow-spiking putative interneurons are shown in purple; broad-spiking pyramidal cells in green. Average waveforms for each class are inset. **(B)** Changes in V4 neurons' spike timing during 5 Hz tACS, color coded by cell type; neurons without significant changes in PLV are shown in black. Numbers near the waveforms indicate the number of cells showing significantly increased, unchanged, and decreased PLVs. The style is otherwise identical to Fig 1A. **(C)** Changes in V4 neurons' spike timing during 20 Hz tACS ($N = 123$), plotted as in panel B. See S1 Data for numeric values. PLV, phase-locking value; tACS, transcranial alternating current stimulation.

overwhelmingly decreased in broad-spiking neurons (88%; 36/41 cells), but the effects on narrow-spiking neurons were more mixed: 12 of 23 narrow-spiking cells became more entrained during 5 Hz tACS, while 11 of 23 showed significantly reduced entrainment. However, this appears to be driven by differences in baseline levels of entrainment: Broad-spiking neurons tended to be strongly entrained at baseline, while narrow-spiking neurons fell into a weakly entrained cluster with PLVs near 0 and a more strongly entrained one with PLVs around 0.3 under baseline conditions. As in the full data set, the effect of tACS on these narrow-spiking neurons was inversely correlated with baseline entrainment levels (Spearman's $\rho = -0.38$; $p = 0.083$). Data collected during 20 Hz tACS, where both cell types showed minimal baseline entrainment, and subsequently became strongly entrained by 20 Hz tACS further supports this hypothesis (Fig 6C). Thus, the effects of tACS appear largely determined by a cell's baseline level of entrainment, rather than its type.

## Discussion

The results shown here demonstrate that tACS can cause individual neurons to fire either more or less rhythmically (Fig 1). The direction of these effects depends both on the relative strengths of entrainment to the ongoing physiological oscillation and the tACS-induced electric field (Figs 2 and 3). These data are consistent with a simple model (Fig 4) in which tACS competes with ongoing oscillations for control of spike timing, a hypothesis supported by several ancillary tests of the model (Figs 5 and 6). Producing predictable changes in neural activity, and, ultimately, behavior, therefore requires detailed knowledge of the relevant oscillations within the targeted brain areas. Our limited understanding of these phenomena, even in the absence of stimulation, may be an underappreciated obstacle to the effective application of tACS. However, once these factors are understood and the stimulation tailored appropriately, the ability to increase and decrease the regularity of spike timing has numerous applications in the laboratory, clinic, and real world.

### Implications for human neuromodulation

The ability to increase and decrease neuronal entrainment opens up a number of new therapeutic and experimental possibilities. Increasing the synchrony of arrhythmically firing neurons by approximately 0.1 PLV may produce meaningful improvements in memory and other cognitive processes (reviewed in 8). Conversely, excess synchrony has been implicated in a wide range of conditions, including movement disorders, epilepsy, and schizophrenia [44]. Parkinson's disease, for example, creates pathological oscillations that entrain neurons with a PLV of up to 0.3 [45,46]. Deep brain stimulation (DBS) has become a standard, albeit invasive and expensive, method for treating severe cases of some of these diseases. While the mechanisms of action are not completely understood, it seems increasingly likely that DBS acts by regulating spike timing in ways that reinstate normal patterns of activity, rather than merely suppressing pathologically strong activity [47]. Our data suggest that tACS may be capable of producing similar effects noninvasively.

The timing of spikes within an oscillatory cycle may also convey information that shapes behavior. In rodents, spike phase relative to ongoing theta oscillations provides detailed information about the position of the animal in space. The existence and role of theta precession are still debated in primates [48,49]; stimulation that shifts spike phase may thus provide a useful tool for investigating its role in human navigation. The phase of spiking also reflects attentional selection, both within [50] and across modalities [51]. These sorts of effects have been studied less frequently in humans, but our data suggest that tACS may also provide a means for doing so.

In power and aeronautical engineering, unwanted oscillations are sometimes controlled by applying open-loop stimulation at a frequency that is slightly "detuned" from that of the original oscillation. This approach, called "asynchronous quenching" or "amplitude death," can be used in any system consisting of self-excited, forced oscillators, including the brain [52]. Asynchronous quenching may therefore provide a simpler and more efficient way to regulate neural activity than the closed-loop systems [1] that are often proposed for this task, simply by applying weak tACS at a frequency slightly below the targeted oscillation (see discussion of asymmetry in the S1 Appendix). Traces of asynchronous quenching also appear in studies using rhythmic sensory stimulation. Listening to a 40 Hz tone suppresses, rather than enhances, gamma activity in auditory cortex [53], while visual flicker can also induce a combination of entrainment and event-related desynchronization [54].

At the same time, these data suggest that it may be much harder than expected to noninvasively increase neuronal entrainment to ongoing oscillations. To do so, stimulation must target brain regions and behavioral conditions in which ongoing entrainment is relatively weak or else it must be precisely matched to both the frequency and phase of the ongoing oscillation. The latter approach is challenging because estimates of frequency and phase provide only transient approximations to the actual oscillations [39], which drift on subsecond timescales [37,38]. Thus, closed-loop stimulation may prove to be more useful for reinforcing existing oscillations.

However, most human tACS use is currently performed open loop, using waveforms of a fixed frequency applied at an arbitrary time. Since those experiments typically seek to "increase," "enhance" (e.g., [55]), or "rescue" [2] an existing neural oscillation, our data raise the worrying possibility that many human tACS experiments have inadvertently produced the exact opposite of their intended effects. For example, membrane potential fluctuations of up to 2 to 3 mV occur in hippocampal neurons [56]. In contrast, the direct polarization produced by tACS generates fluctuations of <1 mV per V/m [25,42]. These ratios therefore correspond to $k = 40\%$ and $80\%$. The precise values may vary between areas and brain states—membrane potential oscillations between UP/DOWN states are much larger, for example—but many human applications seem likely to fall near the border between decreased and increased entrainment.

## Implications for behavioral studies

In that light, we suggest some alternative interpretations of well-established tACS behavioral effects. Detection of faint auditory, visual, and somatosensory stimuli can often be improved by tACS (reviewed in [57]). These results are often ascribed to synchronization between the sensory input and the tACS waveform, as would be expected if tACS further entrained neurons that were already locked to sensory input. However, our data demonstrate that tACS often decreases neural entrainment under these conditions. Interestingly, decreased neural synchrony can sometimes lead to improved sensory performance, as in perceptual decision-making [58,59] or selective attention [60]. Some theories of attention propose that it improves

visual performance by decreasing synchronization at lower frequencies, while simultaneously increasing it at higher ones [61]. Since ongoing low-frequency oscillations tend to be strong, while entrainment to higher frequencies is often weak, a similar pattern of neural effects would seem to be an ideal way for tACS to improve performance on perceptual tasks. Similar mechanisms [62,63] might account for improved sensory discrimination following transcranial magnetic stimulation, as suggested previously [64].

More generally, our results suggest that the effects of tACS likely depend on the behavioral task, insofar as many behaviors modulate the strength of ongoing neural oscillations. Modulating activity in the 8 to 12 Hz "alpha" frequency band is a very common use of tACS, but the dominant frequency of endogenous alpha oscillations varies across subjects due to age, sex, clinical status, and other factors (e.g., [18,19]). Within a single subject, task demands, and overall alertness alter alpha oscillations' amplitude [20,21,65]. As a result, the effects of fixed-frequency tACS at 10 Hz may vary both across subjects and even within a single subject. In fact, because the tACS electric field decays gradually, it may even have opposing effects on different parts of the oscillatory network at the same time. We found opposing effects on neurons recorded 150 μm apart, suggesting that the net effects of tACS may be extremely complicated and determined by both the neurons' physical positions and their functional roles. Phenomena like these may account for some of the extensive intra- and intersubject variability observed in many tACS experiments. Since individual differences in cranial anatomy alter the electric fields produced by tACS, anatomical covariates have been suggested as a way to reduce this variability [11]. Our data suggest that functional covariates, such as the amplitude and peak frequency of the targeted oscillation, measured via electroencephalography (EEG), may be useful as well.

Previous neurophysiological studies, including our own, have not found correlations between baseline activity and tACS's effects on single neurons. We suggest that this is due to Simpson's paradox: The limited range of baseline PLVs in those studies masked the correlations reported here. However, our current findings are consistent with human tES experiments reporting similarly complex nonmonotonic dose–response relationships on behavior [66] and motor-evoked potentials [67]. Our modeling suggests that conditions within the human brain are often near the "elbow" of the curve shown in Fig 4D and that it may be advantageous to try multiple stimulation intensities. Although considerable effort has been devoted to maximizing the field strengths produced by tACS montages, future studies should consider the possibility that, for some goals, more, in terms of current or electric field strength, may not always be better.

## Implications for biophysics

These data challenge previous work suggesting that tACS preferentially engages specific cell types. Based on their elongated, asymmetric morphology, pyramidal cells were predicted to be strongly susceptible to external electric fields while the smaller and bushier interneurons would be much less affected [41,42]. Other work has suggested that network interactions cause tACS to produce especially strong effects in interneurons, which, in turn, drive the remaining neurons [43]. Although these biophysical predictions are likely valid for isolated neurons, they appear less relevant when the neurons are embedded in the complex, interacting networks of a living brain. Neither pattern is evident in our data (Fig 6), where cells' ongoing patterns of activity, rather than morphology or identity, determine how they are affected by tACS. However, neurons' entrainment to ongoing oscillations has been argued to reflect differences in their functional roles, such as their tendency to be affected by top-down input [68]. The inverse correlation we observed may therefore affect these classes differently, and so the

behavioral consequences of tACS may depend on more than the net change in entrainment across the entire neuronal population.

Previous work has also suggested that tACS affects only a small subset of neurons. For example, Johnson and colleagues [7] reported that tACS alters PLVs in 9% to 27% of neurons, depending on the stimulation intensity. Likewise, Ozen and colleagues [6] reported that less than 30% of neurons were entrained by fields of approximately 1 V/m, and our previous work also found tACS-induced PLV changes in only a third to a half of neurons [8,9]. However, these reports considered only individually significant net changes in PLV. The diverse effects demonstrated in Fig 2 suggest that tACS may affect many more neurons by increasing PLV, decreasing PLV, or shifting the phase of entrainment. Thus, experimental constraints and the choice of metrics has led the field to overestimate the sparseness of tACS's effects.

Our experiments focused on relatively low frequencies, which are the dominant target of tACS and persistently occur in vivo. However, some behavioral paradigms can elicit higher-frequency gamma oscillations, and it may be desirable to noninvasively control them with tACS. Simulations suggest that tACS may affect faster oscillations similarly, although the shape of the frequency response curve was markedly broader (S1 Appendix). However, additional experiments are needed to verify these predictions because the underlying biophysics may be different: In particular, it is unclear what happens as the tACS period approaches the membrane time constant, which would be the case for gamma-band stimulation.

Finally, several studies have reported that tACS increases EEG power [69,70], which are often interpreted as evidence of successful neuronal entrainment by tACS. However, EEG signals likely arise from all transmembrane currents [71], including those which do not reach spike threshold. The 2 signals therefore measure slightly different aspects of neural information processing, and so these results are not necessarily incompatible with the ones we report here.

In summary, our results show that at the single neuron level, tACS competes with ongoing brain activity for control of spike timing. The net effect of this competition may be to increase or decrease the overall rhythmicity with which a neuron fires, while shifting the preferred phase of spiking within an oscillatory cycle. As a result, the relatively weak input provided by tACS has the potential to powerfully alter the temporal structure of neural activity.

## Materials and methods

### Experimental design

To characterize the effects of tACS under conditions matching human use, we collected data from 2 adult male rhesus monkeys (*M. mulatta*, 7 and 10 years old; 8 and 14 kg, respectively). The complete data set includes a reanalysis of the experiments described in Vieira and colleagues [9] and Krause and colleagues [8] as well as new data from 157 neurons that have not been reported previously. These new data were collected using approaches and techniques that were virtually identical to our previous work, which are described in detail below.

### Ethics statement

All procedures were approved by the Montreal Neurological Institute's Animal Care Committee (Protocol #5031), followed Canadian Council on Animal Care guidelines, and were supervised by qualified veterinary staff. Animals received regular and varied environmental enrichment, have access to a large play arena, and are socially housed when not in the laboratory, as suggested by the Weatherall Report on the Use of Non-human Primates in Research.

## Behavioral task

We used a simple fixation task to ensure that animals remained in a consistent behavioral state throughout the experiment. Animals sat in a standard primate chair (Crist Instruments, Hagerstown, Maryland, United States of America), placed 57 cm from a computer monitor that covered the central $30° \times 60°$ of the visual field. Eye position was monitored with an infrared eye tracker (SR Research, Ontario, Canada). The monkeys were trained to fixate a small black target (0.5°) presented against a neutral gray background (54 cd/m$^2$) in exchange for juice rewards. Rewards were dispensed according to a schedule with a flat hazard function (exponential distribution with a mean of approximately 1 second). This arrangement minimized external factors known to influence oscillations, such as patterned visual stimulation, eye movements, and predictable rewards, allowing us to examine the effects of tACS against a consistent oscillatory background. Custom software written in MATLAB (The Mathworks, Natick, Massachusetts, USA) controlled the behavioral task and coordinated the eye tracker, tES stimulator, and recording hardware.

## Transcranial alternating current stimulation

We applied tACS in a manner designed to closely mimic human use. To plan the stimulation, individualized finite-element models were built from preoperative MRIs of each animal's head and neck [72]. These were solved to identify electrode configurations that optimally stimulated neurons at our V4 and hippocampal recording sites. The hippocampal configurations produced an average field strength of 0.2 V/m (max: 0.35 V/m) in the hippocampus during± 2 mA stimulation [8]. Since V4 is on the cortical surface, the electric field therein was stronger: approximately 1 V/m at ±1 mA [9]. These conditions therefore bracket the best available estimates for field strengths achievable with 2 electrode montages in humans [73,74]. Our modeling approach also ensured that the brain implants required for these experiment do not produce abnormally strong electric fields elsewhere in the brain.

The stimulation was delivered through 1-cm "high-definition" Ag/AgCl electrodes (PISTIM; Neuroelectrics, Barcelona, Spain). These were coated with a conductive gel (SignaGel; Parker Laboratories, Fairchild, New Jersey, USA) and attached to the animals' intact scalps with a thin layer of Kwik-Sil (World Precision Instruments, Sarasota, Flordia, USA), a biocompatible silicon elastomer. Electrode impedance was typically between 1 and 2 kΩ and always below 10 kΩ to prevent skin damage. In most experiments, current was applied via an unmodified StarStim8 system (Neuroelectrics). The stimulation waveform consisted of a 1.5- or 5-minute sinewave of the specified frequency (5 or 20 Hz) and amplitude (±1 or ±2 mA). At the beginning of each stimulation block, current was linearly ramped up from 0 to the maximum intensity over 5 seconds; this process was repeated in reverse at the end of each block. This minimizes the sensations produced by the onset and offset of stimulation. To measure baseline entrainment, "sham" stimulation repeated the initial and final ramps, without delivering current during the middle of the block. In a few sessions, a DC-STIMULATOR PLUS (NeuroConn, Munich, Germany), driven in remote mode by a Model 4053B signal generator (BK Precision, Yorba Linda, California, USA), was used to deliver stimulation instead. This equipment could not produce the slow ramps, so current was immediately switched on and off instead in these sessions. No significant differences between these sessions in terms of PLV change ($p > 0.31$; Z = 1.01; Wilcoxon rank-sum test) or proportion of neurons affected ($p > 0.1$; $\chi^2$ = 2.83) was observed. Baseline and tACS blocks were separated by an intertrial interval of 1.5 or 5 minutes.

Although stimulation of peripheral afferents in the retina and skin can potentially confound behavioral experiments, we have previously demonstrated that neither likely account for the neural effects reported here [8,9].

## Electrophysiological recording

The target structures were accessed through sterile plastic recording chambers implanted on the skull (Crist Instruments, Hagerstown, Maryland, USA). After penetrating the dura with a 22-gauge stainless steel guide tube, we lowered 32-channel linear arrays (V-Probe; Plexon, Dallas, Texas, USA) into the target with a NAN Microdrive (NAN Instruments, Nazareth Illit, Israel). The location and depths of the targeted structures were confirmed via a postoperative computed tomography (CT) (hippocampus) or MRI (V4) scan.

The raw signals from each electrode were digitized by a Ripple Neural Interface Processor (Ripple Neuro, Salt Lake City, Utah, USA). During acquisition, the signals were band-pass filtered between 0.3 and 7,500 Hz, digitized at 0.5μV/16-bit resolution, and stored at 30,000 Hz for subsequent analysis. The artifacts from tACS stimulation did not exceed the amplifier's linear range (±12 mV), allowing us to continuously record spiking and LFP signals during stimulation. Single units were identified offline by band-pass filtering the signal between 0.5 and 7 kHz with a third-order Butterworth filter. Spikes were initially detected as crossings exceeding a threshold of ±3 standard deviations, robustly estimated for each channel. Short segments around each threshold crossing were then extracted and clustered with UltraMegaSort 2000, a *k*-means based overclustering algorithm [75]. Units were manually reviewed to ensure they had a consistent width and amplitude, a clear refractory period, and good separation in PCA space. Loss of signal during parts of the tACS cycle could produce spurious changes in entrainment, but control analyses for the same equipment and analysis pipeline suggest this is unlikely (e.g., Figure S2 of [8]).

## Phase-locking analysis

We quantified neural entrainment by calculating pairwise phase consistency (PPC) values for each cell, a measure of the synchronization between the phase of an ongoing signal (here, the LFP or tACS) and a point process (spiking activity). Although computationally intensive, this method has several statistical advantages over other common measures of phase locking [76].

These were calculated using spikes obtained from one channel and the continuous signal from an adjacent channel (150 μm away), so as to avoid spectral contamination that may artifactually inflate measures of entrainment [77]. Using this local signal, rather than measuring entrainment to a copy of the tACS output, also ensures that referencing remained constant across conditions and accounts for any physiological distortion of the tACS waveform [78]. To define the phase, the wideband signal was filtered into a ±1 Hz range around the frequency of interest (i.e., 4 to 6 or 19 to 21 Hz). The instantaneous phase at the time of each spike was derived from its Hilbert transform and used to calculate the PPC. PPCs were calculated for each condition separately and compared across conditions via a randomization test. For presentation purposes, these were then converted to PLVs, a more commonly used measure of spike entrainment. Conveniently, PLVs are also equivalent (under some simplifying assumptions) to spike-field coherence, another oft-used metric of spike-LFP coupling.

In the Results section, we focus only on the frequency bands of interest (5 ± 1 or 20 ± 1 Hz) because the effects of tACS were confined to a narrow window around the stimulation frequencies. PLV values calculated in ±1 Hz bands from 2 to 100 Hz revealed significantly increased entrainment only in the 2 to 7 Hz range. Likewise, cells that showed decreased entrainment did not, on average, become coupled to any other frequency component during stimulation (all $p > 0.05$; Wilcoxon sign-rank tests).

Our estimates of oscillation phase were relative to the electric potential in the extracellular space near each neuron. For the baseline condition, this reflects the LFP, while for the tACS condition it is dominated by the instantaneous phase of tACS as detected by our recording

electrode. Thus, the observed differences in the preferred phase of spiking cannot be attributed to a difference in referencing conditions [79], as these were identical for baseline and tACS conditions. Moreover, the change in phase observed with tACS cannot be attributed simply to phase differences between tACS and ongoing oscillations. Given the open-loop nature of our experiments, this difference was random from block to block, but we observed a highly consistent preference change between baseline and tACS blocks.

## Cell type identification

Labeling individual neurons from which data were collected is technically infeasible in nonhuman primates, so putative interneurons were identified based on the shape of their extracellular action potential. Many interneurons contain Kv3 potassium channels, whose fast deactivation rates help quickly repolarize the cell, causing the action potentials to become thinner than those produced by more sluggish channels present in excitatory cells [80]. The trough-to-peak width of the extracellular action potential has therefore been proposed as a method for separating putative inhibitory and excitatory cells [81]. For each neuron, we measured the trough-to-peak width on its average waveform, using spikes collected in all conditions. Our data exhibited a clearly bimodal distribution (Fig 6A), and we applied a hard threshold at 250 μs [82], classifying neurons with narrower spikes as putative interneurons and those with broader spikes as putative excitatory cells. Unsupervised clustering with a Gaussian mixture model yielded very similar classifications.

A more fine-grained categorization of cell types may be possible if additional features of the extracellular action potential are included in the clustering analysis [83]. However, it remains unclear how these functional categories map onto specific morphologies. On the other hand, the 2-way classification used here allows us to test a specific hypothesis about the relationship between cell shape and determine if interneurons are less affected by exogenous electric fields.

However, not every narrow-spiking neuron in our data set is likely to be a fast-spiking GABAeric interneuron. Previous work has identified neurons that emit narrow action potentials despite having spiny, pyramidal morphology [84] and, in primates, up to 25% of Kv3-expressing neurons are non-GABAergic [85]. An alternate explanation for our findings in Fig 6 is that these non-GABAergic narrow-spiking neurons are affected by tACS, while the "true" interneurons are unresponsive. However, the proportion of affected cells in our data (23/51) is significantly higher than would be expected under a 25%/75% split ($p < 0.01$; $\chi^2(1) = 10.99$), suggesting that this is unlikely to occur in our data.

## Model simulations

The Stuart–Landau oscillator described above consists of 2 ordinary differential equations with 3 parameters [35]. The frequency of the ongoing oscillation is determined by the ω parameter, while its stability is controlled by λ: Values above zero produce self-sustaining oscillations, while setting $\lambda < 0$ causes the oscillations to decay with time. Finally, γ dampens the system, controlling how quickly it settles into a steady-state amplitude.

The model equations were solved numerically using MATLAB's `ode45` function, an explicit Runge–Kutta method [4,5], using the solver's default parameters and the initial conditions $x = 0$, $y = -1$. The model was run for 200 seconds total. In stimulation conditions, the external drive s(t) was included only after 40 seconds so that the model could reach steady state. In both cases, we defined the amplitude as $\sqrt{2}$ times root–mean–squared amplitude of the signal; this corresponds to an amplitude of 1 for a standard sine wave but corrects for any transients created by numerical integration. To generate Fig 4C, we evaluated all combinations of phase mismatch (0 to 315˚, in steps of 45˚) and frequency mismatch (50% to 150% of the

baseline oscillation's cycle, in steps of 10%). We tested values of $k$ ranging from 0% to 100% of the baseline oscillation's amplitude, a range that corresponds to physiologically relevant conditions (See Discussion, "Implications for human neuromodulation"). These were then numerically integrated with MATLAB's `trapz` to find the net effect across phase and frequency shown in Fig 4D. Similar results were obtained using Julia and its `Tsit5` solver.

As the model does not have explicit physical units, we performed a limited exploration of the parameter space, to determine whether a qualitative match to our data could be found. For Fig 4, the values were $\lambda = 0.2$, $\omega = 0.5$, and $\gamma = 1.0$, which were similar to those used in a previous study of the model [35]. However, these precise parameters were not required to reproduce the negative flanks shown in Fig 4C and 4D. We ran 250 additional simulations, setting $\lambda$, $\omega$, and $\gamma$ independently to random values between 0.1 and 10; smaller values took prohibitively long to reach steady state. In more than half of these simulations (162/250), at least one value of $k$ (10–150%, in steps of 10%) reduced the total entrainment, as in Fig 4D. Indeed, the "quenching" behavior we have observed is actually a universal feature of many kinds of oscillators [86], and as shown in S1 Appendix, similar results can be derived analytically from a simplified version of this model [87].

## Quantification and statistical analysis

Ethical and practical considerations limit the number of animals from which we can collect data. However, the critical comparisons in this paper are made within (e.g., tACS versus baseline) and between (e.g., putative pyramidal versus interneuron) individual neurons, making the cell rather than the animal the relevant unit of analysis [88]. Where possible, statistical analyses were performed using nonparametric tests that avoid distributional assumptions; population-level analyses were carried using Wilcoxon rank-sum and sign-rank tests, as appropriate. We used 95% CIs of the median as population dispersion, calculated using the formula in [89]. All statistical tests are 2 tailed, using sample sizes derived from our previous work. Significance of differences between categories was assessed using the $\chi^2$ test. Sample sizes were determined based on our previous work, and data were analyzed using MATLAB (The Mathworks), the CircStats toolbox [90], and Julia [91]. As we reviewed in [8], the effect sizes reported here are similar to those produced by changes in behavioral state and sensory input and, therefore, are likely to be physiologically relevant.

## Supporting information

**S1 Fig. Baseline entrainment of neurons in each experimental condition. (A)** Entrainment spectra for V4 neurons ($N = 157$), showing phase locking to the LFP in $\pm$ 1 Hz frequency bands. The median and 95% CI are shown, calculated from all V4 neurons (i.e., those used in subsequent 5 Hz and 20 Hz tACS experiments). Red arrows indicate the tACS frequencies used in those experiments. **(B)** Entrainment spectra for hippocampal neurons ($N = 21$), plotted in the same style. **(C)** Individual baseline values for each cell in the 3 experiment conditions. The red cross indicates the median and $^{**}$ indicates significance at the $p < 0.01$ level. See also Fig 1; note that the V4 data are divided across panels A and B of Fig 1. Numeric values can be found in S1 Data. LFP, local field potential; tACS, transcranial alternating current stimulation. (TIFF)

**S2 Fig. Both model populations are affected similarly by tACS.** The values of $x$ (black) and $y$ (magenta) during 3 runs of the model with $k = 5\%$, 30%, and 75%, demonstrating that they are phase-shifted copies of each other. Note that baseline condition is not shown here, but is

shown in Fig 4. tACS, transcranial alternating current stimulation.
(TIFF)

**S1 Appendix. Additional discussion and mathematical analysis of the oscillator model.**
(DOCX)

**S1 Data. Numeric values for each data point in Figs 1–6 and S1 Fig.**
(XLSX)

**S2 Data. Numeric values for each data point in Fig 4.** See also the code contained in the repository linked from the Data Availability Statement.
(XLSX)

# Acknowledgments

We thank Julie Coursol, Cathy Hunt, and Dr. Fernando Chaurand for outstanding technical assistance. Melanie Segado provided helpful comments on the paper.

# Author Contributions

**Conceptualization:** Matthew R. Krause, Pedro G. Vieira, Christopher C. Pack.

**Formal analysis:** Matthew R. Krause, Pedro G. Vieira, Jean-Philippe Thivierge, Christopher C. Pack.

**Funding acquisition:** Matthew R. Krause, Christopher C. Pack.

**Investigation:** Matthew R. Krause, Pedro G. Vieira.

**Methodology:** Matthew R. Krause, Pedro G. Vieira, Christopher C. Pack.

**Supervision:** Christopher C. Pack.

**Visualization:** Matthew R. Krause, Pedro G. Vieira.

**Writing – original draft:** Matthew R. Krause, Pedro G. Vieira, Jean-Philippe Thivierge, Christopher C. Pack.

**Writing – review & editing:** Matthew R. Krause, Pedro G. Vieira, Jean-Philippe Thivierge, Christopher C. Pack.

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
