## [Editor Report · Decision Letter 0]

19 Jan 2022

Dear Dr Krause, 

Thank you for submitting your manuscript entitled "Brain stimulation competes with ongoing oscillations for control of spike timing in the primate brain" for consideration as a Research Article by PLOS Biology.

Your manuscript has now been evaluated by the PLOS Biology editorial staff, as well as by an academic editor with relevant expertise, and I am writing to let you know that we would like to send your submission out for external peer review.

Once your full submission is complete, your paper will undergo a series of checks in preparation for peer review. Once your manuscript has passed the checks it will be sent out for review. To provide the metadata for your submission, please Login to Editorial Manager (https://www.editorialmanager.com/pbiology) within two working days, i.e. by Jan 21 2022 11:59PM.

If your manuscript has been previously reviewed at another journal, PLOS Biology is willing to work with those reviews in order to avoid re-starting the process. Submission of the previous reviews is entirely optional and our ability to use them effectively will depend on the willingness of the previous journal to confirm the content of the reports and share the reviewer identities. Please note that we reserve the right to invite additional reviewers if we consider that additional/independent reviewers are needed, although we aim to avoid this as far as possible. In our experience, working with previous reviews does save time. 

If you would like to send previous reviewer reports to us, please email me at ggasque@plos.org to let me know, including the name of the previous journal and the manuscript ID the study was given, as well as attaching a point-by-point response to reviewers that details how you have or plan to address the reviewers' concerns. 

Given the disruptions resulting from the ongoing COVID-19 pandemic, please expect some delays in the editorial process. We apologise in advance for any inconvenience caused and will do our best to minimize impact as far as possible.

Kind regards,

Gabriel

Gabriel Gasque

Senior Editor

PLOS Biology

ggasque@plos.org

---

## [Decision Letter · Decision Letter 1]

11 Feb 2022

Dear Dr Krause,

Thank you for submitting your manuscript "Brain stimulation competes with ongoing oscillations for control of spike timing in the primate brain" for consideration as a Research Article at PLOS Biology. Your manuscript has been evaluated by the PLOS Biology editors, by an Academic Editor with relevant expertise, and by three independent reviewers.

In light of the reviews (below), we will not be able to accept the current version of the manuscript, but we would welcome re-submission of a much-revised version that takes into account the reviewers' comments. We cannot make any decision about publication until we have seen the revised manuscript and your response to the reviewers' comments. Your revised manuscript is also likely to be sent for further evaluation by the reviewers.

We expect to receive your revised manuscript within 3 months. 

**IMPORTANT - SUBMITTING YOUR REVISION**

Your revisions should address the specific points made by each reviewer. We recommend that you keep the modelling part but try to clarify to address reviewer 3's concerns. You should also take the reviewers' recommendations regarding more analyses and discussion.

Please submit the following files along with your revised manuscript:

*Re-submission Checklist*

*Published Peer Review*

*PLOS Data Policy*

*Blot and Gel Data Policy*

Sincerely,

Gabriel

Gabriel Gasque

Senior Editor

PLOS Biology

ggasque@plos.org

REVIEWS:

Reviewer #1, Robert M. G. Reinhart: Summary

Transcranial alternating current stimulation (tACS) has been widely used to noninvasively modulate brain function via neural entrainment of targeted regions at a specific frequency. However, like other noninvasive brain stimulation methods, tACS receives criticisms because of its variable effects. In this article, the authors provided a promising explanation for the inconsistent effects, that is, tACS competes with the brain's ongoing oscillations. They computed phase-locking values (PLVs) that describe the consistency of spike timing and compared PLV during stimulation to baseline. The results showed that tACS modulation interacts with current brain states and successful entrainment happens when the intrinsic rhythm is weak. When neurons are strongly locked to ongoing activity, tACS works via desynchronization and re-entrainment. Model simulations demonstrated that frequency mismatch between the intrinsic oscillation and tACS stimulation is the primary cause of entrainment decrease.

I think these are novel and fascinating findings. However, some clarifications and discussion are needed to strengthen the interpretation as well as to elucidate the mechanism. Below are the concerns that should be addressed.

Major

Results section, tACS alters the strength and phase of entrainment. 

1. Overall, tACS altered neural activity in two dimensions: entrainment as measured by bidirectional PLV changes, and shifts of the preferred phase. Could the author discuss what's the functional relevance and difference of the two changes in terms of tACS effects on behavior?

2. Phase shift analysis was performed on 1) neurons without PLV changes but showed phase preference and 2) all neurons including those with no significant PLV changes. Both results suggested the occurrence of phase change irrespective of tACS entrainment. Could the authors explain the specific motivation to perform the same analysis on the sub-portion neurons without significant PLV changes? If these analyses speak to the same conclusion, why not report one of the two results and put the other one in the supplementary?

3. The two analyses suggest a similar phase shift toward 35.6 and 38. However, there was no explanation of why it falls in this range. Was it driven by the tACS stimulation starting phase? Second paragraph of this section (11th line) writes 'the rising phase of the tACS waveform'. Also see the last sentence in the second paragraph of Results section 'tACS first desynchronizes, then re-entrains neural activity'. Does the rising phase of tACS waveform suggest the starting phase of tACS stimulation? If yes, providing this information in Methods will solve the confusion. 

4. S2 demonstrates the phase shift of individual neurons who had PLV increase or decrease during tACS stimulation. However, there was no information on the average phase change. Is it possible to draw a red arrow like the one in Figure 2. This will provide the opportunity to compare whether the phase shift of neurons with PLV change differs from the phase shift of neurons without PLV changes.

5. The authors introduced one novel finding of the present study in the final paragraph of the section based on S2 (tACS first decouples neural activity and then re-entrains them). I am not sure why the authors decided to put these plots in supplementary, rather than the main text. I think it might be more important than the example abcd neurons in Figure 2.

Results section 'tACS first desynchronizes, then re-entrains neural activity'

1. The authors' interpretation of Figure 3C is that tACS competes with ongoing oscillations and exerts influence via desynchronization and re-entrain. This account seems to view the desynchronization and re-entrain process as two temporally ordered steps. I would reconsider using 'first-then' because this temporal relationship is based on independent conditions (baseline, 1mA tACS, and 2mA tACS). I am not sure why the 1mA condition (entrainment elimination) has to be the intermediate state of baseline and 2mA condition. In other words, whether desynchronization is a prerequisite for neurons with strong intrinsic oscillation when receiving intensive tACS stimulation (e.g., 2mA)? Considering the result of model simulation in Figure 4C and how baseline frequency affects tACS effects in Figure 5, I wonder if it's possible that the entrainment decrease in 1mA tACS condition was mainly driven by frequency mismatch. The frequency mismatch issue also exists in the 2mA tACS condition, but high intensity stimulation has a larger tolerance window of frequency mismatch as shown in Figure 4C. This would raise the possibility to observe successful entrainment in the 2mA condition reflected as PLV increase in Figure 3C. Meanwhile, how can the authors defend against an alternative explanation of Figure 3C, that is, once the tACS intensity reaches the threshold to evoke entrainment, ongoing brain activity will be immediately tuned into targeted frequency.

2. In the last paragraph of this section, the authors mentioned 'the preferred phase for these neurons was similar to that observed in V4'. Could the author extend the discussion on why both populations showed converged preferred phase around 0 as shown in Figure 3C and 3D? 

Results section 'Similar patterns of effects emerge from a simple oscillator model'

1. In this section, the author examined the mechanism that underlies entrainment decrease and found that frequency mismatch between tACS and ongoing oscillation accounts for the findings. Figure 4C and 4D are informative demonstrations of model simulation results. I consider the relative entrainment change on the y-axis is obtained by filtering LFP at the stimulated frequency (e.g., 5Hz) and then comparing PLV between baseline and tACS condition. If this is true, I am confused why tACS would compete with ongoing oscillation at the same frequency and desynchronize it. 

2. How to interpret the asymmetry of Figure 4C that same frequency mismatch on the left side actually had a stronger effect than those on the right side?

3. Was entrainment change of y-axis in Figure 4C and 4D measured by amplitude (power) change or PLV change? Axis units are missing in both plots. I wonder whether y-axes in 4C and 4D used same scales?

4. Color bar in Figure 4C indicates tACS amplitude relative to baseline which is the parameter k in the first equation. This parameter determines whether one will have an entrainment increase or decrease at a given stimulation frequency (Figure 4D). Is it possible to calculate the corresponding k of 1mA and 2mA relative to baseline in the current study?

Minor

Figure 1A and 1B showed PLV change of example neurons. Having four examples in A but only two in B might produce the impression that tACS entrainment increment is more dominant. However, the median PLV actually decreased in tACS condition relative to baseline. If the purpose of using more examples in Figure 1A is to demonstrate the phase shift, specify it in the caption will be helpful.

Title Figure 1C, 1D, and 1E will make it more straightforward that they represent populations from different regions and in different frequency bands.

Figure 1D and 1E are not referred to in the main text. I believe Results section 'Decreased entrainment is not caused by stimulation frequency or brain region' is relevant to the two plots.

Figure 6, adding legend in 6B and 6C will make it easier to understand that purple dots represent interneurons. Again, title 6B and 6C will help to understand they demonstrate results of 5Hz and 20Hz tACS respectively. Maybe it's common in electrophysiology but the broad neuroscience audience might get confused by the subpanel in Figure 6A and 6B. Further descriptions either in the caption or in Methods are recommended.

Reviewer #2: Combining experimental and theoretical methods, the authors have studied the compound effect of tACS on modulating brain activity and they provided an interesting and convincing explanation for the variability of tACS effects. They found that a competition between the external tACS modulation and the internal ongoing oscillations leads to variable or even opposite effects of tACS modulation, which depends on both the strength and frequency of the tACS modulation relative to ongoing oscillations. The study is important and timely, given that tACS is a widely-used noninvasive method for modulating brain activity and behavior but its effects were not reliable or even contradictory. This study gave a clear picture for the tACS effects which were not well understood yet. The experiments, model simulation and data analysis were all carefully done and the paper was well written. There is no major concern from me on this paper, but I suggest the authors to consider the followings for further discussion to support their conclusions. I strongly support the acceptance of this paper.

1) It is nice that the authors showed the competition between internal oscillations and external tACS regarding to their relative frequency and strength (Figure 4). It will be even better to show the final effect as a function of relative phases between the external and internal oscillations. If the internal brain oscillation is regular in frequency, then I can roughly predict the final modulatory effect depending on the relative phase; but when the brain oscillation is irregular, as several studies have shown, the phase effect might be also complicated with the consideration of oscillation frequency. The author should have some discussion about the effect from relative phases between internal and external signals. Some further simulations with the current model regarding to oscillation phases might be better (for example systematically change or randomly perturb phases of external signals).

2) The competing effects were well demonstrated in this paper with a biology plausible model for brain oscillations at low frequency (such as theta, alpha or even beta), which is corresponding to their experimental data collected at 5Hz and 20Hz. I suggest to add more discussion for brain modulations at high frequency. At the conceptual level, I don't think the conclusion in this paper will be changed, but in details there might be different form of such a competition and phase-locking. When tACS goes to the modulation at high frequency (above 40Hz or in gamma band), the competing effect and its cell specificity might be different, because studies have shown that underlying mechanisms or dynamic models (for example, Jia et al. 2013 and Han et al. 2021) for brain oscillation at high frequency were different from what was used in the paper for low-frequency oscillations. I understand that the authors might not have experimental data with tACS stimulation at high frequency, but they should give some insights on what will happened when brain oscillations were generated in a different model in the discussion. This also holds for cell-specificity, because for gamma oscillations, excitatory and inhibitory neurons were coupled more tightly, but they might not be coupled (as the model in the paper showed) for brain oscillations at low frequency.

Reviewer #3: This is an important study because addresses some sources of possible variability to tACS effects in humans, even touching some issues that have been not considered up to now. Results come out from two adult male rhesus monkeys: I am not an expert of monkey studies, so maybe this is a sufficient number of animals. Methodology is sound, and already published in relevant Journals.

However, as it stands the manuscript is extremely long and, overall, a bit hard to read even for a specialist of the field. One possibility to reduce this problem is to take off all the modeling part, that has by itself the potential for a separate (theoretical) work. In this way, moreover, there is no risk to interpret modelling results (for example those on tACS effects on different cell types) as consequence of the physiological experiment.

One issue that is not properly addressed and should be better conceptualized and discussed is the relationship between spike activity of a single neuron (i.e., the main focus of the current study), the overall firing rate (whose changes have been never observed in this study, according to Fig. 1 and text) and the complex oscillatory networks that are the target of human tACS studies. This seems a crucial point for fully appreciating the potential translational relevance of the observed effects. I presume, for example, that a change of spiking in a small subset of neurons belonging to a discrete cortical region could be entirely shadowed (either "electrically" and, of course, behaviorally) within a distributed oscillatory activity of a complex network.

In Fig. 2 I would modify the colors of the baseline and tACS dots: now they appear as blu and black, respectively, and it is not so easy to distinguish them in the figure.

---

## [Decision Letter · Decision Letter 2]

13 Apr 2022

Dear Dr Krause,

Thank you for resubmitting your manuscript "Brain stimulation competes with ongoing oscillations for control of spike timing in the primate brain" for consideration as a Research Article by PLOS Biology. Your study was re-evaluated by two of the independent reviewers and, I am happy to say, they are supportive of moving forward with this work.

Before we can formally accept this manuscript for publication, we ask that you make sure to address all of our data and other policy-related requests (see the bottom of this email). We also ask that you consider a potentially different title for this work that more clearly conveys the use of direct recordings in this work and that it provides and explanation for why brain stimulation isn't always enhancing. You could consider something along the lines of "Neural entrainment to ongoing oscillations reverses the enhancing effects of external brain stimulation"...though I appreciate that this specific title implies that stimulation was thought always be enhancing. We are not wedded to the need for a title change but ask that you consider how to most clearly convey your interesting findings to our broad audience that includes non-cognitive neuroscientists and non-neuroscientists as well.

As you address the various items, please take this last chance to review your reference list to ensure that it is complete and correct. If you have cited papers that have been retracted, please include the rationale for doing so in the manuscript text, or remove these references and replace them with relevant current references. Any changes to the reference list should be mentioned in the cover letter that accompanies your revised manuscript.

We expect to receive your revised manuscript within two weeks. 

*Published Peer Review History*

*Press*

Sincerely,

Kris

Kris Dickson,

Neurosciences Senior Editor/Section Manager,

kdickson@plos.org,

PLOS Biology

ETHICS STATEMENT:

-- Please include the specific national or international regulations/guidelines to which your animal care and use protocol adhered. Please note that institutional or accreditation organization guidelines (such as AAALAC) do not meet this requirement.

The requirements for non-human primate studies for publication in PLOS Biology are at:

https://journals.plos.org/plosbiology/s/animal-research#loc-non-human-primates

And they state:

Non-human primate studies must be performed in accordance with the recommendations of the Weatherall report “The use of non-human primates in research”. Manuscripts describing research involving non-human primates must include details of animal welfare, including information about housing, feeding, and environmental enrichment, and steps taken to minimize suffering, including use of anesthesia and method of sacrifice if appropriate.

DATA POLICY:

We appreciate that your data has already been made available at https://osf.io/9t2yp/?view_only=f678fff106274d67aaa65cbb36bf5c75. 

Please double check to ensure that you provide the individual numerical values that underlie the summary data displayed in the relevant figure panels as they are essential for readers to assess your analysis and to reproduce it. NOTE: the numerical data provided should include all replicates AND the way in which the plotted mean and errors were derived (it should not present only the mean/average values).

In addition to providing access to the individual numerical values that underlie the summary data, please also ensure that figure legends IN YOUR MANUSCRIPT include information on WHERE THE UNDERLYING DATA CAN BE FOUND, and ensure that your supplemental data file/s has a legend that allows readers to clearly associate the data in your study with the deposited raw data. This is needed for all relevant figures (Fig1A-E; Fig2A-C; Fig3A-D; Fig4C-D; Fig5A-B; Fig6A-C; SuppFigS1A-C).

DATA NOT SHOWN?

Reviewer remarks:

Reviewer's Responses to Questions

PLOS authors have the option to publish the peer review history of their article (what does this mean?). If published, this will include your full peer review and any attached files.

Reviewer #1: No

Reviewer #3: No

Reviewer #1: The authors have satisfactorily addressed all of my concerns. This will make a fine contribution to the field. 

-Rob Reinhart

Reviewer #3: I congratulate with the Authors for the quality and balancing of their response to reviewers' comments.

The manuscript can be published, on my side

---

## [Editor Report · Decision Letter 3]

27 Apr 2022

Dear Dr Krause,

On behalf of my colleagues and the Academic Editor, Huan Luo, I am pleased to say that we can in principle accept your Research Article "Brain stimulation competes with ongoing oscillations for control of spike timing in the primate brain" for publication in PLOS Biology, provided you address any remaining formatting and reporting issues. These will be detailed in an email from our production team that will follow this letter and that you will usually receive within 2-3 business days, during which time no action is required from you. Please note that we will not be able to formally accept your manuscript and schedule it for publication until you have completed any of their requested changes.

PRESS

Sincerely, 

Kris

Kris Dickson 

Senior Editor 

PLOS Biology

kdickson@plos.org